# Beyond Reward Margin: Rethinking and Resolving Likelihood Displacement in Diffusion Models via Video Generation

## Abstract

Direct Preference Optimization (DPO) has shown promising results in aligning generative outputs with human preferences by distinguishing between chosen and rejected samples. However, a critical limitation of DPO is **likelihood displacement**, where the probabilities of chosen samples paradoxically decrease during training, undermining the quality of generation. Although this issue has been investigated in autoregressive models, its impact within diffusion-based models remains largely unexplored. This gap leads to suboptimal performance in tasks involving video generation. To address this, we conduct a formal analysis of DPO loss through **updating policy within the diffusion framework**, which describes how the updating of specific training samples influences the model's predictions on other samples. Using this tool, we identify two main failure modes: (1) Optimization Conflict, which arises from small reward margins between chosen and rejected samples, and (2) Suboptimal Maximization, caused by large reward margins. Informed by these insights, we introduce a novel solution named **Policy-Guided DPO** (PG-DPO), combining Adaptive Rejection Scaling (ARS) and Implicit Preference Regularization (IPR) to effectively mitigate likelihood displacement. Experiments show that PG-DPO outperforms existing methods in both quantitative metrics and qualitative evaluations, offering a robust solution for improving preference alignment in video generation tasks.

## 1 Introduction

Generative diffusion models (Wan et al. (2025); Tan et al. (2025); Gao et al. (2025); Kong et al. (2024)) have made remarkable progress in video generation. However, aligning these models with human preferences for visual quality and text-to-video fidelity remains a significant challenge, limiting their practical applicability. Although generative models have benefited from model-based alignment methods such as RLHF (Ouyang et al. (2022); Xu et al. (2023)) and GRPO (Shao et al. (2024); Xue et al. (2025); Liu et al. (2025a)), their application to video generation is severely hampered due to the reliance on explicit pre-trained reward models. The high dimensionality and temporal complexity of video make the development of a comprehensive reward model exceptionally difficult. Therefore, the scarcity of effective reward models (Pu et al. (2024)), combined with extreme computational demands (Li et al. (2025)), makes the exploration of efficient, model-free alignment techniques not only beneficial but critical for advancing video generation.

To address these challenges, Direct Preference Optimization (DPO, Rafailov et al. (2024)) emerges as a compelling alternative, learning preferences directly from paired data without an explicit reward model. By reframing alignment as a direct policy optimization problem, DPO bypasses the costly reward modeling stage and has yielded promising results in diverse domains (Pal et al. (2024); Yang et al. (2024a); Black et al. (2024); Wallace et al. (2023)). However, a critical challenge in applying DPO is **likelihood displacement** (Pal et al. (2024); Razin et al. (2025); Tajwar et al. (2024)), where the probabilities of chosen samples paradoxically decrease during training, leading to a degradation of the overall quality and diversity of generation. As illustrated in Fig. 2a, this issue is readily observed in the training of existing methods like VideoDPO Liu et al. (2024). Although previous works have explored solutions for likelihood displacement in autoregressive language models (Razin et al. (2025); Yang et al. (2025); Ren & Sutherland (2025)), these approaches lack a formal analysis

of the underlying mechanism in diffusion models, let alone a solution tailored to high-dimensional data such as video.

This paper fills this gap by conducting a formal analysis of DPO's **updating dynamics** within diffusion framework. We decompose the change in the model's prediction into three distinct terms that contribute differently to the model's overall behavior. This framework is general and can be adapted to various fine-tuning algorithms, including supervised finetuning (SFT, Wei et al. (2022)), DPO and other modern preference optimization methods (Zhao et al. (2022); Pal et al. (2024); Azar et al. (2023), shown in Appendix A.3.2). Our analysis pinpoints two primary scenarios that cause likelihood displacement: (1) **Optimization Conflict** with Small Reward Margins: When the reward margin is extremely small, our analysis reveals that for such pairs, the updating policy between chosen and rejected samples are nearly the same, leading to an optimization conflict that suppresses the likelihood of both samples rather than effectively separating them; (2) **Suboptimal Maximization** with Large Reward Margins: When the reward margin is already large, the strength of updating policy tends to vanish, preventing the model from further increasing the probability of high-quality chosen samples. We argue that this stagnation is itself a form of likelihood displacement, as the model fails to progressively allocate more probability mass to superior samples.

These insights motivate the development of a novel solution: **Policy-Guided DPO** (PG-DPO), an algorithm designed to resolve both failure modes of likelihood displacement within the diffusion framework. PG-DPO comprises two key components: (1) Adaptive Rejection Scaling (ARS): To resolve the optimization conflict, ARS introduces an adaptive weight $\alpha \in [0, 1)$ to the reward of the rejected sample in the Bradley-Terry model(BT, Sun et al. (2025)). This dampens the "push" signal from rejected samples, breaking the harmful optimization similarity. As a result, the "pull" signal from chosen samples dominates the update, ensuring a net possibility increase instead of joint suppression. (2) Implicit Preference Regularization (IPR): To overcome suboptimal maximization, IPR introduces a regularization term focused solely on the chosen sample. This regularizer ensures that a consistent "pull" signal is always present, encouraging the model to continuously enhance the likelihood of chosen samples, even when their reward margin is already substantial.

In summary, our main contributions are as follows:

1. We present the first formal analysis of DPO's updating policy in diffusion models, attributing the likelihood displacement problem to two fundamental failure modes: optimization conflict and suboptimal maximization.

2. This framework not only offers a unified perspective for various fine-tuning methods, but also inspires a novel algorithm, PG-DPO, which synergizes ARS and IPR to specifically counteract these two failure modes.

3. Experiments show that PG-DPO not only mitigates likelihood displacement but also achieves state-of-the-art performance in preference alignment for video generation, confirmed by both quantitative metrics and qualitative evaluations.

## 2 RELATED WORK

**Video Diffusion Models** Video generation aims to produce visually coherent videos that align with user intent, with applications ranging from story animation He et al. (2023), interactive game development (Che et al. (2024)), to embodied artificial intelligence (Chen et al. (2024b)). Initial models, such as VDM (Ho et al. (2022b)), adapted 2D U-Net architectures for spatio-temporal modeling but were limited by high computational costs and low resolution. Consequently, the field has largely shifted to Transformer-based architectures, often by extending powerful text-to-image backbones with temporal attention mechanisms (Zheng et al. (2024); Peng et al. (2025); Yang et al. (2024b)). Models based on the Diffusion Transformer (DiT) architecture now represent the state-of-the-art (Wan et al. (2025); Kong et al. (2024); Gao et al. (2025)), achieving significant gains in generation quality and consistency (Chen et al.; Esser et al. (2023); Ruan et al. (2023)). Despite this progress, aligning these powerful models with nuanced human preferences remains a key unresolved challenge (Wu et al. (2021); Ho et al. (2022a)).

**Aligning Generative Models with Human Preferences** Aligning generative models with human preferences originated in the development of Large Language Models (LLMs). The seminal ap-

proach, Reinforcement Learning from Human Feedback (RLHF, Ouyang et al. (2022)), utilizes a two-stage process: training an explicit reward model and then optimizing the policy using reinforcement learning. Later, Direct Preference Optimization (DPO, Rafailov et al. (2024)) greatly simplified this pipeline by reframing alignment as a direct supervised learning problem on preference pairs. This formulation avoids both explicit reward modeling and the instabilities of RL training. DPO's simplicity and effectiveness have since spurred a family of related methods in natural language processing aimed at improving feedback efficiency and scalability (Lai et al. (2024); Meng et al. (2024); Swamy et al. (2024); Zhao et al. (2022); Yuan et al. (2023)).

**Aligning Diffusion Models with Human Preferences** These alignment principles have been adapted to diffusion models to enhance aesthetic quality and text faithfulness. Existing approaches fall into three main paradigms. The first directly adapts DPO-style objectives; for instance, Diffusion-DPO (Wallace et al. (2023)) formulates a DPO loss using the Evidence Lower Bound (ELBO) as a log-likelihood proxy. The second uses policy gradients akin to classic RLHF, as seen in DDPO (Black et al. (2024)) and DPOK (Fan et al. (2023)), but often inherits its training instability. The third and most common paradigm relies on an explicit reward model, either through direct gradient backpropagation (e.g., DRaFT Clark et al. (2024), VADER Prabhudesai et al. (2024)) or more advanced optimization schemes (e.g., FlowGRPO Liu et al. (2025a), DanceGRPO Xue et al. (2025)). Notably, a fourth, emerging paradigm is unsupervised alignment, exemplified by SPIN-Diffusion (Yuan et al. (2024)), which uses a self-play mechanism to generate preference data, thereby reducing the dependency on human annotation.

## 3 PG-DPO: Discovery, Formulation and analysis

### 3.1 Preliminary

**Diffusion Model.** Stable Diffusion (SD, Rombach et al. (2022)) is a latent diffusion model that maps a random noise vector $z_t$ and a text prompt $\mathcal{P}$ to an output video $I_0$, aligning with the given conditioning prompt via cross-attention. The objective of this process is defined as:

$$\min_\theta \mathbf{E}_{z_0, \epsilon \sim N(0,I), t \sim Uniform(1,T)} \|\epsilon - \epsilon_\theta(z_t, t, c_{\text{text}}, c_{\text{img}})\|_2^2, \tag{1}$$

where $c_{\text{img}}$ denotes the reference image, $c_{\text{text}}$ denotes the text condition, $t \in [0, T]$ denotes the time step in the diffusion process. $\epsilon$ and $\epsilon_\theta$ represent the actual and predicted noise, respectively. The noise is gradually removed by sequentially predicting it using pre-trained diffusion model.

**Reinforcement learning from human feedback (RLHF).** RLHF (Ouyang et al. (2022)) trains the model by maximizing the reward of the output of the model, while regularizing the KL-divergence between it and reference model. Specifically, RLHF trains a reward function $r(c, x)$ that estimates the human preference on the generated sample $x$ conditioned on $c$. Assume $p_\theta$ denotes the optimizing model, $p_{ref}$ denotes the reference model and the hyperparameter $\beta$ controls the strength of KL regularization. The objective of RLHF is calculated as:

$$\max_\theta \mathbf{E}_{c,x}[r(c, x)] - \beta \mathbf{D}_{KL}[p_\theta(x|c)\|p_{\text{ref}}(x|c)]. \tag{2}$$

**Direct preference optimization (DPO).** Direct preference optimization (Rafailov et al. (2024)) simplifies RLHF by training the model directly from human preferences. The objective of DPO $\mathcal{L}_{dpo}$ is defined as below:

$$-\min_\theta \left[ \log \sigma \left( \beta T \log \left( \frac{p_\theta(x_{t-1}^w | x_t^w, c)}{p_{ref}(x_{t-1}^w | x_t^w, c)} \right) - \beta T \log \left( \frac{p_\theta(x_{t-1}^l | x_t^l, c)}{p_{ref}(x_{t-1}^l | x_t^l, c)} \right) \right) \right], \tag{3}$$

where $x_t^w$ and $x_t^l$ denote the "chosen" and "rejected" samples generated from the condition $c$ in time step $t$, $\sigma(\cdot)$ denotes the sigmoid function.

### 3.2 Updating Policy of Finetune Algorithm

Updating policy refers to how changes in specific samples influence the model's output. When the model updates its parameters using gradient descent (GD), there are:

$$\Delta\theta = \theta^{\tau+1} - \theta^\tau = -\eta \cdot \nabla\mathcal{L}_{DPO}(x^w, x^l), \tag{4}$$

where the update of $\theta$ during step $\tau \to \tau + 1$ is given by one gradient update on the sample pair $(x^w, x^l)$ with learning rate $\eta$. The updating policy in this paper answers the question: *After one GD on $(x^w, x^l)$, how does the model's prediction on any $x_0$ change?*

The origin updating policy might be the "stiffness" (Fort et al. (2020)) or "local elasticity" (He & Su (2020); Deng et al. (2021)) of neural networks. See Appendix A.3.2 for more discussion. Given that updating policies have been successfully applied to many deep learning systems, yielding remarkable benefits, we extend this framework to diffusion models. The updating policy in timestep $t$ of Eq. 4 become:

$$\Delta \log p_{\theta^{\tau+1}}(x_{t-1}^o | x_t^o) = \log p_{\theta^{\tau+1}}(x_{t-1}^o | x_t^o) - \log p_{\theta^\tau}(x_{t-1}^o | x_t^o). \tag{5}$$

To ensure the generality of our analysis, we investigate the updating policy of diffusion models under the Markov assumption. The generalizability of this analysis under other models (Esser et al. (2024); Song et al. (2021)) will be discussed in Appendix A.3.2.

Let $p_\theta(x_{t-1} | x_t) \sim \mathcal{N}\left(\mu_\theta(x_t), \sigma_t^2 I\right)$, the one-step updating policy can be decomposed as:

$$\Delta \log p_{\theta^{\tau+1}}(x_{t-1}^o | x_t^o) \propto \eta T \beta (1-a) \mathcal{G}_\theta(x_t^o) \cdot (\mathcal{K}_\theta(x_t^o, x_t^w) \mathcal{G}_\theta^T(x_t^w) - \\ \mathcal{K}_\theta(x_t^o, x_t^l) \mathcal{G}_\theta^T(x_t^l)) + \mathcal{O}(\eta^2 \|\nabla \mathcal{L}_{DPO}\|^2). \tag{6}$$

where $\mathcal{G}_\theta(x_t^o) = \Sigma^{-1}(x_t^o - \mu_\theta(x_t^o))$, $\mathcal{K}_\theta(x_t^o, x_t^w) = \langle \nabla_\theta \mu(x_t^o), \nabla_\theta \mu(x_t^w) \rangle$, $\mathcal{K}_\theta(x_t^o, x_t^w)$ is the empirical neural tangent kernel of the logit network $\mu_\theta$, and more details can be seen in Appendix A.3.2.

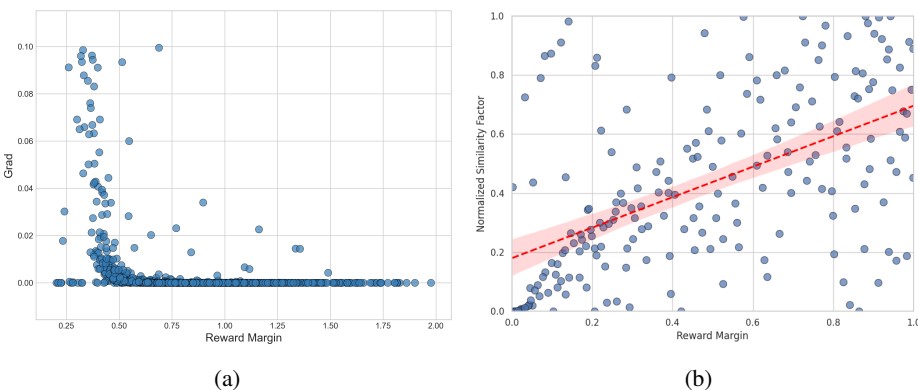

(a)                                    (b)

Figure 1: **Empirical Validation of DPO's Failure Modes.** (a) Suboptimal Maximization: The gradient norm of DPO vanishes for large reward margins, stifling further optimization of chosen samples as the learning signal for "easy" pairs disappears. (b) Optimization Conflict: A strong positive correlation between reward margin and similarity factor (defined as $\Delta \log p_{\theta^{\tau+1}}(x_{t-1}^w | x_t^w) - \Delta \log p_{\theta^{\tau+1}}(x_{t-1}^l | x_t^l)$). Here, a low factor leads to nearly identical updates for chosen and rejected samples, causing their optimization signals to interfere.

Next, we analyze the likelihood displacement based on updating policy during DPO training, where the probability of chosen responses can paradoxically decrease. We define the core term within the sigmoid function of DPO loss as the **reward margin** (detailed definition shown in Eq. 30). Our analysis reveals that the likelihood displacement problem stems from two critical failure modes, occurring at opposite ends of the reward margin.

**Suboptimal Maximization with Large Margins**: This failure mode arises when the model effectively distinguishes a preference pair $(x^w, x^l)$, resulting in a large reward margin. As empirically validated in Fig. 1a, the DPO gradient norm plummets towards zero precisely as the reward margin grows. According to Eq. 29, the larger margin leads to a larger $a$, causing update magnitude to vanish. This vanishing gradient indicates that the learning signal for these "easy" pairs disappears, disincentivizing the model from further increasing the probability of even exemplary chosen samples. Consequently, the learning process stagnates prematurely, leading to a suboptimal policy that fails to fully capitalize on strong preference signals.

**Optimization Conflict with Small Margins**: Conversely, a more critical failure mode emerges at small reward margins, where the model struggles to differentiate the pair. DPO is designed to apply its maximum corrective force in this regime. However, herein lies a conflict rooted in the inherent dynamics of diffusion models. Fig. 1b reveals this conflict by showing a strong positive correlation between the reward margin and an update similarity factor. At small margins, this factor is also minimal, implying that the updates for the chosen ($x^w$) and rejected ($x^l$) samples become nearly identical. This means that precisely when DPO tries hardest to enforce separation, the model's update mechanism treats both samples almost interchangeably. The aggressive update intended to suppress the rejected sample's probability inadvertently "spills over" and penalizes the similar chosen sample. This struggle creates a direct optimization conflict, causing likelihood displacement for the chosen response and degrading model quality.

Crucially, this analysis of failure modes is agnostic to the origins of $x^w$ and $x^l$. This makes our framework broadly applicable to other DPO-style algorithms (Azar et al. (2023); Zhao et al. (2022)), whose primary distinction lies in how they formulate the reward margin. More details on this extension are provided in Appendix A.3.2.

### 3.3 PG-DPO

Motivated by our analysis of the DPO update policy, we introduce PG-DPO, a novel alignment algorithm. PG-DPO integrates two key components: Adaptive Rejection Scaling (ARS) and Implicit Preference Regularization (IPR), allowing for more stable and effective preference alignment.

**Adaptive Rejection Scaling (ARS)**   ARS is designed specifically to resolve the optimization conflict inherent in DPO. The core idea is to dynamically adjust the penalty applied to the rejected sample ($x^l$) based on the reward margin. A smaller margin, indicating a difficult-to-distinguish pair, should receive a gentler penalty to avoid penalizing the chosen sample ($x^w$) as well. To implement this, we introduce an adaptive weight, $\alpha(x^w, x^l)$, which modulates the repulsive gradient from the rejected sample:

$$\alpha(x_t^w, x_t^l) = \sigma\left[ K_1 \cdot \frac{r_t^w - r_t^l}{r_t^l + \epsilon} \right], \tag{7}$$

where $r_t$ represents the implicit reward $\log(p_\theta / p_{ref})$, $K_1$ is a scaling hyperparameter, $\epsilon$ is a small constant for numerical stability. The sigmoid function $\sigma(\cdot)$ normalizes the weight to prevent high variance and ensure stable training.

The mechanism of ARS directly addresses the aforementioned conflict. As previously established, this conflict is most severe for low-margin pairs, where DPO's maximal gradient objective causes the suppressive gradient from $x^l$ to inadvertently "leak" and penalize the similar chosen sample $x^w$. ARS resolves this via its adaptive weight $\alpha$: when the reward margin becomes very small, the numerator in Eq. 7 approaches zero, which effectively down-weighting the rejected sample's contribution. Rather than aggressively separating a nearly indistinguishable pair, the optimization pressure from $x^l$ is automatically moderated. This reduction prevents the strong negative update from suppressing the probability of $x^w$. In essence, ARS gracefully shifts the optimization focus for low-margin pairs. It moves away from punishing the rejected sample and instead prioritizes reinforcing the chosen one. By acting as a dynamic safeguard, ARS ensures that the model does not paradoxically degrade good samples when faced with ambiguous preference data, thereby robustly preventing likelihood displacement.

**Implicit Reference Regularization (IPR)**   IPR addresses the issue of Suboptimal Maximization. This problem occurs when DPO's learning stagnates for high-margin pairs as the optimization gradient vanishes. IPR is designed to ensure continuous improvement for chosen samples ($x^w$) even in these well-separated cases. To achieve this, we introduce $\gamma(x_t^w, x_t^l)$, which is designed to activate for large margins. The weight is calculated as follows, where $K_2$ is a scaling hyperparameter:

$$\gamma(x_t^w, x_t^l) = \sigma\left[ -K_2 \cdot \left( \frac{r_t^w - r_t^l}{r_t^l + \epsilon} \right) \right]. \tag{8}$$

IPR's mechanism creates a smooth transition between two optimization objectives. As the reward margin ($r_t^w - r_t^l$) grows large, the input to the sigmoid function in Eq. 8 becomes a large negative

number, causing $\gamma$ to approach 0. This process smoothly deactivates the standard DPO margin-maximization objective and simultaneously activates a secondary objective focused solely on the chosen sample.

Crucially, this secondary objective is not a naive SFT loss, but rather maximizes the implicit reward $r_t^w = \log\left(p_\theta(x_t^w|c)/p_{\text{ref}}(x_t^w|c)\right)$. This "reference-regularized" formulation encourages the model to increase the likelihood of exemplary samples while remaining anchored to the reference model. Therefore, IPR ensures continued improvement without sacrificing model stability.

**Full Objective:** By integrating ARS and IPR, we arrive at the final objective of PG-DPO:

$$
\mathcal{L}_{\text{ours}} = -\min\left[\gamma(x_t^w, x_t^l)\log\sigma\left(\beta T\log\left(\frac{p_\theta(x_{t-1}^w|x_t^w, c)}{p_{\text{ref}}(x_{t-1}^w|x_t^w, c)}\right)\right.\right.
$$
$$
\left.\left. - \alpha(x_t^w, x_t^l)\beta T\log\left(\frac{p_\theta(x_{t-1}^l|x_t^l, c)}{p_{\text{ref}}(x_{t-1}^l|x_t^l, c)}\right)\right) + (1 - \gamma(x_t^w, x_t^l))\log\left(\frac{p_\theta(x_{t-1}^w|x_t^w, c)}{p_{\text{ref}}(x_{t-1}^w|x_t^w, c)}\right)\right].
$$
(9)

This composite objective provides a comprehensive solution to DPO's failure modes. For any preference pair $(x^w, x^l)$, the training dynamic is governed by the reward margin:

- **Small Margin:** When the margin is small, $\gamma$ is close to 1 and $\alpha$ become smaller than 1. The loss is dominated by the ARS-modified DPO objective, which softens the penalty on $x^l$. This prevents the Optimization Conflict problem.
- **Large Margin:** When the margin is large, $\gamma$ approaches 0. The loss smoothly transitions to the IPR objective (maximizing $r_\theta(x^w)$). This provides a direct and persistent signal to improve the probability of chosen sample, overcoming the Suboptimal Maximization problem.

In essence, our method acts as an intelligent controller for the optimization process. It applies a softened DPO objective for ambiguous pairs and switches to a regularized reinforcement objective for clearly decided pairs. This dual mechanism ensures robust and continuous learning across the entire spectrum of preference data, directly resolving the core failure modes of DPO.

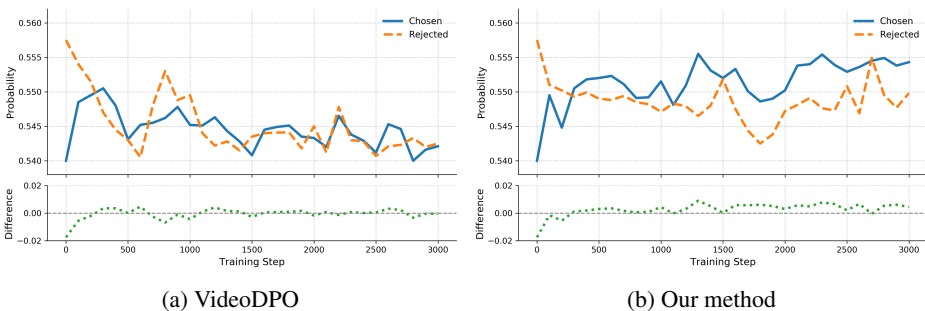

(a) VideoDPO                    (b) Our method

Figure 2: **Predicted probabilities for chosen and rejected samples across training steps.** (a) VideoDPO suffers from likelihood displacement. (b) Our PG-DPO maintains a consistent upward trend for chosen samples, demonstrating its ability to resolve the problem.

**Analysis** We analyze the effects of PG-DPO based on the probability of chosen samples $\log p_\theta(x_{t-1}^w|c)$, where $c$ denotes the input conditions of diffusion model. This reflects the model's ability to align with human preference by raising the generated probability of chosen samples. We define the target function as follows:

$$
\Gamma(\theta, \tau + 1) = \log p_{\theta^{\tau+1}}^{\text{Ours}}(x_t^w|c) - \log p_{\theta^{\tau+1}}^{\text{VideoDPO}}(x_t^w|c)
$$
$$
= -\eta\nabla_{\theta^\tau}\log p_{\theta^\tau}(x_t^w|c)\cdot(\nabla_{\theta^\tau}^T\mathcal{L}_{\text{ours}} - \nabla_{\theta^\tau}^T\mathcal{L}_{\text{VideoDPO}}).
$$
(10)

To empirically validate PG-DPO, we analyzed its ability to consistently enhance the probabilities of chosen samples. We track the per-step log-likelihood advantage, denoted by $\Gamma(\theta, \tau + 1)$, which

measures the additional increase in probability for chosen samples compared to the VideoDPO (supported by Appendix A.4). Crucially, our experiments show this advantage is positive in the majority of training steps: **81.7%** for the Wanx2.1-14B model Wan et al. (2025) and **76.9%** for VideoCrafter-v2 Chen et al. (2024a). This provides strong quantitative evidence that PG-DPO effectively mitigates likelihood displacement by robustly and consistently boosting the probabilities of desired responses.

As illustrated in Fig. 2, which tracks sample probabilities on VC2 model during training, we further present the effectiveness of our method. VideoDPO Liu et al. (2024) exhibits the likelihood displacement, where the probabilities of both chosen and rejected samples degrade over time. In stark contrast, our approach not only achieves a stable increase in the reward margin but also ensures the chosen sample's probability grows consistently, demonstrating a healthier alignment process (more details can be seen in Appendix A.5.2).

## 4 EXPERIMENTS

### 4.1 SETUP

**Baseline.** We compare our method with several open-source models for video generation: VideoCrafter-v2 (VC2, Chen et al. (2024a)) for text-to-video generation and Wanx2.1-14B (Wanx2.1, Wan et al. (2025)) for image-to-video generation. These models are utilized as baselines in our alignment experiments. We compare our method with baseline model, VideoDPO (Liu et al. (2024)) and SFT (Wei et al. (2022)) results, the traditional finetuning methods and the model-free reinforcement method within video generation.

**Metric.** To evaluate our method and the baselines, we adopt six metrics (CLIP-score Hessel et al. (2022), HPS-v2 Wu et al. (2023), VideoAlign Liu et al. (2025b), Temporal flickering Teed & Deng (2020), Aesthetic Quality LAION-AI. (2022), VQA scores Wu et al. (2022) and a user study), covering both non-human and human preference metrics (details can be seen in Appendix A.5.1).

**Training Details.** We train all the models for 3000 steps with batch size 16, using the Adam optimizer with a learning rate of $1 \times 10^{-6}$. The reweighted hyperparameters are set to $K_1 = 10, K_2 = 10$ and $\beta = 100$ (supported by Appendix A.5.3). For each prompt, the number of generated videos is set to 4. The highest- and lowest-scoring videos, as determined by the overall score of VideoAlign, are respectively labeled as the chosen and rejected samples. This procedure results in 3000 training video pairs. All experiments are conducted on 32 NVIDIA H20 GPUs.

### 4.2 MAIN RESULTS

**Quantitative Comparison.** We present our quantitative results in Tab. 1, evaluating two state-of-the-art open-source video models: the UNet-based model (VC2) and the DiT-based model (Wanx2.1). After training with our method, both models demonstrate marked performance improvements, showing consistent gains across all evaluated metrics. Compared to Supervised Fine-Tuning (SFT), which utilizes only chosen samples within the dataset, our method yields superior results in both semantic and visual quality. This highlights the effectiveness of PG-DPO, particularly in improving model performance with limited data. Furthermore, models trained with VideoDPO show erratic performance, sometimes even regressing below the baseline model. This underscores the robustness and reliability of our optimization strategy. In summary, the consistent enhancements across both UNet-based and DiT-based architectures affirm the broad applicability and generalizability of our proposed method within video generation tasks.

**Qualitative Comparison.** Fig. 3 and Fig. 4 present qualitative comparisons, showcasing our method's consistent improvements in both visual fidelity and prompt alignment. For instance, our method enables Wanx2.1-14B to generate a more coherent spatial layout between the food and chopsticks (Fig. 4). Similarly, when applied to VC2, it produces a more realistic and visually appealing subject. In terms of text-video alignment, our method strikes a superior balance between adhering to textual constraints, such as the "hold a balloon" prompt (Fig. 3), and maintaining overall visual quality. Across all examples, our method yields generations superior to competing approaches and more results are shown in the Appendix A.6.

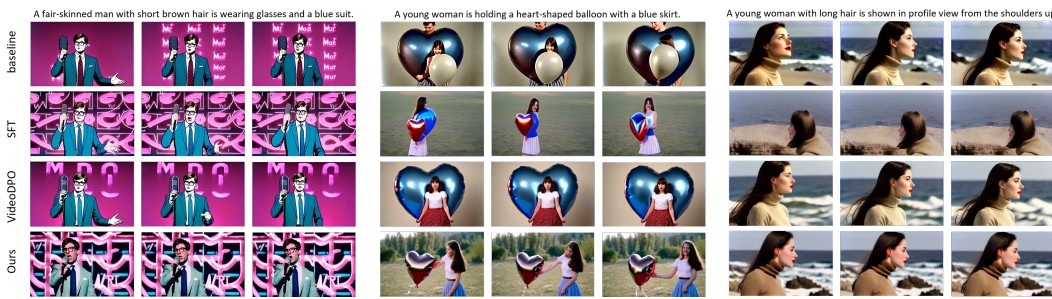

Figure 3: Comparison results with previous works on the VideoCrafter (VC2).

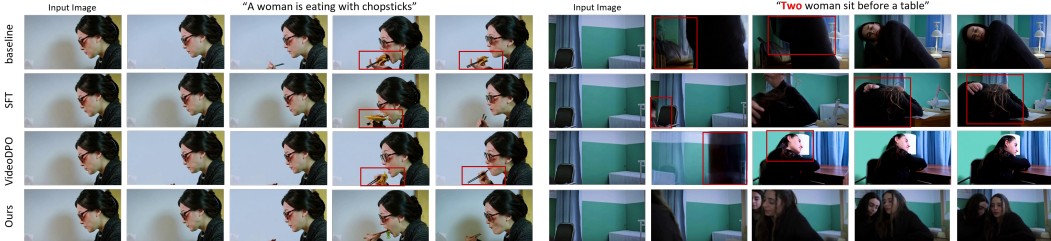

Figure 4: Comparison results with previous works on the Wanx2.1.

Table 1: Quantitative comparisons with previous works on Wanx2.1 and VideoCrafter2 (VC2).

| Model | Method | clip↑ | HPS↑ | VideoAlign | | | Temporal flickering↑ | Aesthetic Quality↑ | VQA↑ |
|---|---|---|---|---|---|---|---|---|---|
| | | | | VQ↑ | MQ↑ | TA↑ | | | |
| Wanx2.1 | baseline | 0.2898 | 0.2378 | 0.1601 | 0.8280 | 0.4592 | 0.9745 | 0.6729 | 0.8681 |
| | SFT | 0.2998 | 0.2314 | 0.2603 | 0.8769 | 0.5518 | 0.9827 | 0.6610 | 0.8668 |
| | VideoDPO | 0.2952 | 0.2333 | 0.2124 | 0.8551 | 0.4826 | 0.9810 | 0.6829 | 0.8709 |
| | Ours | **0.3156** | **0.2551** | **0.2937** | **1.0231** | **0.6265** | **0.9919** | **0.6985** | **0.8869** |
| VC2 | baseline | 0.2576 | 0.2057 | -0.1723 | 0.5027 | 0.1379 | 0.9935 | 0.5469 | 0.7717 |
| | SFT | 0.2815 | 0.2143 | -0.8116 | 0.0966 | 0.4462 | 0.9919 | 0.5301 | 0.7444 |
| | VideoDPO | 0.2631 | 0.2334 | -0.0345 | 0.7034 | 0.1555 | 0.9990 | 0.5511 | 0.7842 |
| | Ours | **0.2947** | **0.2494** | **-0.0012** | **0.9949** | **0.5239** | **0.9997** | **0.5722** | **0.8009** |

## 4.3 ABLATION ANALYSIS

To isolate the contributions of each component within PG-DPO, we evaluate the performance of PG-DPO against several variants: (1) VideoDPO (denoted as baseline); (2) PG-DPO with IPR; and (3) PG-DPO with ARS. The results are summarized in Tab. 2.

Table 2: Ablation study on the Wanx2.1 and VideoCrafter2 (VC2) models. We compare the PG-DPO (Ours) against the baseline (VideoDPO) and two variants adding only ARS or IPR.

| Model | Method | clip↑ | HPS↑ | VideoAlign | | | Temporal flickering↑ | Aesthetic Quality↑ | VQA↑ |
|---|---|---|---|---|---|---|---|---|---|
| | | | | VQ↑ | MQ↑ | TA↑ | | | |
| Wanx2.1 | baseline | 0.2952 | 0.2333 | 0.2124 | 0.8551 | 0.4826 | 0.9810 | 0.6829 | 0.8709 |
| | +ARS | 0.2997 | 0.2427 | 0.2237 | 1.0195 | 0.4969 | 0.9844 | 0.6839 | 0.8745 |
| | +IPR | 0.3078 | 0.2376 | 0.2747 | 0.8770 | 0.6183 | 0.9818 | 0.6846 | 0.8850 |
| | Ours | **0.3156** | **0.2551** | **0.2937** | **1.0231** | **0.6265** | **0.9919** | **0.6985** | **0.8869** |
| VC2 | baseline | 0.2576 | 0.2057 | -0.1723 | 0.5027 | 0.1379 | 0.9935 | 0.5469 | 0.7717 |
| | +ARS | 0.2714 | 0.2465 | -0.1573 | 0.7986 | 0.3392 | 0.9988 | 0.5568 | 0.7894 |
| | +IPR | 0.2796 | 0.2213 | -0.1685 | 0.5972 | 0.3040 | 0.9965 | 0.5643 | 0.7939 |
| | Ours | **0.2947** | **0.2494** | **-0.0012** | **0.9949** | **0.5239** | **0.9997** | **0.5722** | **0.8009** |

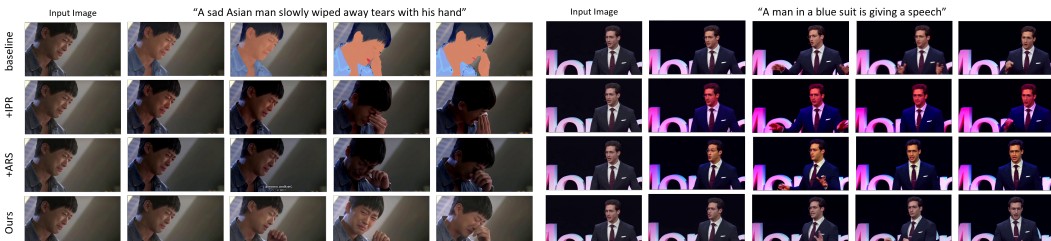

Figure 5: Ablation results on the Wanx2.1.

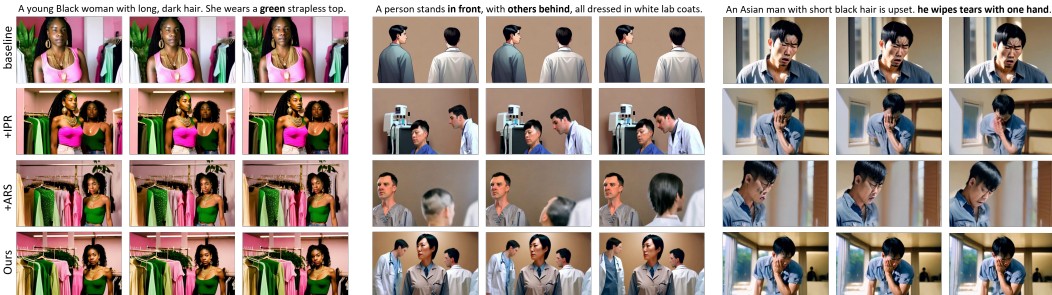

Figure 6: Ablation results on the VideoCrafter (VC2).

**Effect of ARS.** We validate the effectiveness of our ARS module by comparing results with and without it. As illustrated in Fig. 5, ARS mitigates the generation degradation that can be caused by the likelihood-shift effect in VideoDPO. Furthermore, as shown in the first and second cases in Fig. 6, ARS enhances the model's text-video alignment, enabling it to correctly generate video content for prompts like "a woman in a green dress" and "a person standing in front." Quantitative evaluations in Tab. 2 confirm these benefits, showing that our method with ARS outperforms the baseline on all metrics, with particularly significant enhancements in MQ and temporal flickering.

**Effect of IPR.** We demonstrate the critical role of IPR by comparing results with and without this module. As shown in our qualitative and quantitative results (Fig. 5, Fig. 6 and Tab. 2), IPR yields substantial improvements over the baseline. For instance, the third case in Fig. 6 shows that with IPR, the generated video more accurately depicts the description "wipe tears with one hand". Videos generated with IPR also exhibit superior visual quality, which is corroborated by the significant improvements in metrics such as VQ, Aesthetic Quality, and VQA.

## 5 CONCLUSION

In this paper, we conduct a formal analysis of the **updating policy** for fine-tuning diffusion models, which is a theoretical framework that is broadly applicable for understanding the behavior of various alignment algorithms. Through this lens, we identified two root causes of likelihood displacement, the phenomenon responsible for DPO's critical challenges: optimization conflicts on low-margin data and suboptimal maximization on high-margin data. These insights directly motivated the design of our novel algorithm, PG-DPO. By integrating Adaptive Rejection Scaling (ARS) to resolve optimization conflicts and Implicit Preference Regularization (IPR) to ensure continuous improvement on high-quality samples, PG-DPO systematically addresses both failure modes. Our empirical results on both U-Net and DiT-based video generation models demonstrate that PG-DPO achieves consistent and significant improvements over existing fine-tuning methods in both quantitative metrics and qualitative evaluations. In conclusion, the analysis of the updating policy offers a unified framework for interpreting and diagnosing issues within diffusion model fine-tuning. The resulting algorithm, PG-DPO, not only validates this theoretical foundation but also provides a robust and generalizable solution for aligning diffusion-based video generation models with human preferences.

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

# A APPENDIX

## A.1 DISCUSS, FUTURE WORK AND ACKNOWLEDGEMENTS.

**Discussion** We introduce an updating policy framework that unifies and interprets preference alignment methods by deconstructing their loss functions into core update dynamics. This framework reveals an inherent asymmetry in the DPO update mechanism, which, when combined with the similarity effect, provides a principled explanation for the "likelihood displacement" failure mode. Building on this diagnosis, we propose PG-DPO, which introduces two controllable parameters, $\alpha$ and $\gamma$, to explicitly manage the update strength trade-off and apply adaptive regularization to chosen samples. The empirical success of PG-DPO not only resolves DPO's instabilities but also validates our framework's diagnostic power, advocating for a more principled, mechanics-driven approach to designing alignment algorithms.

**Limitation and Future Work** A primary limitation is that PG-DPO introduces two hyperparameters, $K_1$ and $K_2$, which require careful tuning. Future work will focus on developing adaptive mechanisms to automate their selection, making the algorithm more robust and user-friendly. Furthermore, our current framework is centered on DPO-style methods. A key direction is to extend it to encompass RL-based alignment techniques like PPO, aiming to bridge the gap between these paradigms. Finally, we plan to leverage the framework not only to investigate its interaction with varying data quality but also as a blueprint for designing novel alignment algorithms from first principles.

**Acknowledgements** We thank Gemini for its assistance in refining the language of this manuscript. We also appreciate the anonymous reviewers for their constructive comments, which will help improve this work.

## A.2 USE OF LLMS

The Large Language Models (LLMs) are utilized for language polishing and proofreading of this manuscript. Assistance from the LLMs are not sought for the conceptual ideas or the code. The authors assume full responsibility for any and all issues that may arise from its use.

## A.3 UPDATING POLICY OF FINETUNE ALGORITHM

In this section, we provide a detailed derivation and analysis of the Updating Policies for various fine-tuning methods within the diffusion framework. The goal is to foster a deeper understanding of their underlying update mechanisms and to offer insights into potential failure modes during the optimization process. Specifically, we want to emphasize a critical distinction between our methodology and traditional analyses that focus solely on the gradient of the loss function. While the gradient indicates the magnitude or intensity of learning in a given scenario, our analysis of the Updating Policy directly reveals how the probability of a sample changes following a single update step. This approach, therefore, offers a more direct and interpretable view of the model's dynamic behavior.

### A.3.1 INSTRUCTION FINETUNING USING SUPERVISED FINE-TUNING (SFT)

Suppose that we want to monitor how the model's prediction changes on an "observing example" $x^o$ after a single fine-tuning step. Starting from Equation 5, we first approximate $\log p_{\theta^{\tau+1}}(x^o_{t-1}|x^o_t, c^o)$ via a first-order Taylor expansion:

$$
\begin{aligned}
\log p_{\theta^{\tau+1}}(x^o_{t-1}|x^o_t, c^o) = {} & \log p_{\theta^\tau}(x^o_{t-1}|x^o_t, c^o) + \\
& \langle \nabla \log p_{\theta^\tau}(x^o_{t-1}|x^o_t, c^o), \theta^{t+1} - \theta^\tau \rangle + O(\|\theta^{\tau+1} - \theta^\tau\|^2),
\end{aligned}
\tag{11}
$$

where $\tau$ denotes the training step, $x^o_t$ denoted the noisy latent embedding of $x^o$ in timestep $t$. Then, assuming the model updates its parameters using SGD calculated by an "updating example" $x^u$ with condition $c^u$, we can rearrange the terms in the above equation to get the following expression:

$$\Delta logp_{\theta^{\tau+1}}(x_{t-1}^o|x_t^o, c^o) = logp_{\theta^{\tau+1}}(x_{t-1}^o|x_t^o, c^o) - logp_{\theta^\tau}(x_{t-1}^o|x_t^o, c^o)$$
$$= \nabla_\theta \log p_{\theta^\tau}(x_{t-1}^o|x_t^o, c^o)(\theta^{\tau+1} - \theta^\tau) + O(\|\theta^{\tau+1} - \theta^\tau\|^2). \tag{12}$$

To evaluate the leading term, we plug in the definition of SGD:

$$\Delta logp_{\theta^{t+1}}(x_{t-1}^o|x_t^o, c^o) = -\eta\nabla_\theta logp_{\theta^t}(x_{t-1}^o|x_t^o, c^o)(\nabla_\theta\mathcal{L}_{\text{SFT}}(x^u)|_{\theta^\tau})^T$$
$$\propto -\eta\, \mathcal{G}_\theta(x_t^o) \cdot \mathcal{K}_\theta(x_t^o, x_t^u) \cdot \mathcal{G}_\theta^T(x_t^u) + O(\eta^2), \tag{13}$$

where $\eta$ denotes the learning rate, denote $\mathcal{K}$ and $\mathcal{G}$ as $\mathcal{G}_\theta(x_t^o) = \Sigma^{-1}(x_t^o - \mu_\theta(x_t^o))$, $\mathcal{K}_\theta(x_t^o, x_t^w) = \langle\nabla_\theta\mu(x_t^o), \nabla_\theta\mu(x_t^w)\rangle$. Next, we prove Eq. 13 under two main forms of diffusion model. First, as for DDPM schedule Ho et al. (2020), the posterior relationship between the noisy samples $x_t^o$ and $x_{t-1}^o$ conditioned on $c^o$ is:

$$p_\theta(x_{t-1}|x_t) \sim \mathcal{N}\left(\sqrt{\overline{\alpha}_{t-1}}\hat{x}_0 + \sqrt{1 - \overline{\alpha}_{t-1} - \sigma_t^2} \cdot \frac{x_t - \sqrt{\overline{\alpha}_t}\hat{x}_0}{\sqrt{1 - \overline{\alpha}_t}}, \sigma_t^2 I\right), \tag{14}$$

where,

$$\hat{x}_0 = \frac{x_t - \sqrt{1 - \overline{\alpha}_t}\hat{\epsilon}_t(x_t, t)}{\sqrt{\overline{\alpha}_t}}. \tag{15}$$

As for Flow matching Esser et al. (2024); Liu et al. (2025a), the posterior relationship between $x_t^o$ and $x_{t-1}^o$ conditioned on $c^o$ is as follows:

$$dx_t = (v_\theta(x_t, t) + \frac{\sigma_t^2}{2t}(x_t + (1 - t)v_\theta(x_t, t)))dt + \sigma_t dw_t, \tag{16}$$

where $dw_t$ denotes Wiener process increments and $\sigma_t$ controls the stochasticity strength, $v_\theta(\cdot)$ denotes the learned denoise models, as denoted as $\epsilon_\theta$ in DDPM schedule. To simulate the stochastic flow matching trajectory, the Eq. 16 can be discretized via Euler-Maruyama method:

$$x_{t+\Delta t} = x_t + (v_\theta(x_t, t) + \frac{\sigma_t^2}{2t}(x_t + (1 - t)v_\theta(x_t, t)))\Delta t + \sigma_t\sqrt{\Delta t}\epsilon, \tag{17}$$

where $\sigma_t = a\sqrt{t/1 - t}$ and $a$ is a scalar hyper-parameter that controls the noise level, $\epsilon \sim \mathcal{N}(0, I)$, $t \in [0, 1]$ denotes the denoise step, $\Delta t$ is the integration step.

According to above analysis, we can set $p_\theta(x_{t-1}|x_t) \sim \mathcal{N}\left(\mu_\theta(x_t), \sigma_t^2 I\right)$ for simplify under these two main forms. According to Maximum Likelihood Estimation (MLE), we calculate the SFT loss:

$$\mathcal{L}_{\text{SFT}} = -\log p_\theta(x_{t-1}|x_t)$$
$$= \log\left((2\pi)^{-\frac{m}{2}}\Sigma^{-\frac{m}{2}}\exp\left(-\frac{1}{2}(x_t - \mu_\theta(x_t))^T\Sigma^{-1}(x_t - \mu_\theta(x_t))\right)\right) \tag{18}$$
$$\propto \min\left((x_t - \mu_\theta(x_t))^T\Sigma^{-1}(x_t - \mu_\theta(x_t))\right),$$

where $m$ denotes as the dimension of $x_t$. As a result, we calculate the gradient of SFT loss:

$$\nabla_\theta\mathcal{L}_{\text{SFT}} = -\nabla_\theta\log p_\theta(x_{t-1}|x_t, c) = -\frac{\partial\log p_\theta(x_{t-1}|x_t, c)}{\partial\mu_\theta(x_t, c)} \cdot \frac{\partial\mu_\theta(x_t, c)}{\partial\theta} \tag{19}$$
$$\propto -\left((\Sigma^{-1}x_t - \Sigma^{-1}\mu_\theta(x_t, c)) \cdot \nabla_\theta\mu_\theta(x_t, c)\right).$$

Plugging Eq. 19 into Eq. 13, which yields the SFT fine-tuning update policy:

$$\Delta \log p_{\theta^{t+1}}(x_{t-1}^o|x_t^o, c^o) = -\eta \nabla_\theta \log p_{\theta^t}(x_{t-1}^o|x_t^o, c^o)(\nabla_\theta \mathcal{L}_{\text{SFT}}(x^u)|_{\theta^\tau})^T + O(\eta^2)$$

$$\propto -\eta((\Sigma^{-1}x_t^o - \Sigma^{-1}\mu_\theta(x_t, c)) \cdot \nabla_\theta \mu_\theta(x_t^o, c^o))((\Sigma^{-1}x_t^u - \Sigma^{-1}\mu_\theta(x_t^u, c^u)) \cdot \nabla_\theta \mu_\theta(x_t^u, c^u))^T$$

$$= -\eta \, \mathcal{G}_\theta(x_t^o) \cdot \mathcal{K}_\theta(x_t^o, x_t^u) \cdot \mathcal{G}_\theta^T(x_t^u). \tag{20}$$

The second term in this decomposition, $\mathcal{K}(x_t^u, x_t^o)$, is the product of gradients at $x_t^o$ and $x_t^u$. This matrix is known as the empirical Neural Tangent Kernel (NTK), which can be interpreted as the network's learned similarity metric that evolves throughout training. Intuitively, the Frobenius norm of this kernel is large when the gradients point in similar directions. For appropriately initialized wide networks trained with a small learning rate, prior work (Arora et al. (2019); Jacot et al. (2018)) has shown that $\mathcal{K}$ remains nearly constant, and its magnitude is larger for more similar sample pairs. This theoretical property provides a formal basis for the intuitive behavior observed in Supervised Fine-Tuning (SFT): an update step on a sample $x^u$ induces a more pronounced change in the probability of other, more similar samples. We also observe a parallel "similarity effect" in the DPO loss update mechanism, as shown in Sec. A.3.2. As illustrated in Fig. 1b, when the reward margin is small, the differential impact on the probabilities of various samples diminishes, with their respective changes converging towards zero.

### A.3.2 INSTRUCTION FINETUNING USING DPO-STYLE FINE-TUNING

Direct Preference Optimization (DPO, Rafailov et al. (2024)) is usually considered the first model-free alignment algorithm for preference fine-tuning. Hence, we start from DPO and other DPO-style fine-tuning algorithms to analyze the updating policy.

The training loss of DPO on the example preference pairs $(x^w, x^l)$ conditioned on $c$ denotes:

$$\mathcal{L}_{dpo} = -\min\left[\log \sigma\left(\beta T \log\left(\frac{p_\theta(x_{t-1}^w|x_t^w, c)}{p_{ref}(x_{t-1}^w|x_t^w, c)}\right) - \beta T \log\left(\frac{p_\theta(x_{t-1}^l|x_t^l, c)}{p_{ref}(x_{t-1}^l|x_t^l, c)}\right)\right)\right]. \tag{21}$$

where $T$ denotes the total denoise step, $\beta$ denotes the strength of KL diversity. Similar to Equation 13 for SFT, the updating policy for the DPO loss can be written as:

$$\Delta \log p_{\theta^{t+1}}(x_{t-1}^o|x_t^o, c^o) = -\eta \nabla_\theta \log p_{\theta^t}(x_{t-1}^o|x_t^o, c^o)(\nabla_\theta \mathcal{L}_{\text{DPO}}(x^w, x^l)|_{\theta^\tau})^T. \tag{22}$$

To get the updating policy of DPO, we first calculate the gradient of DPO loss and repeatedly use the chain rule:

$$\begin{aligned} \nabla_\theta \mathcal{L}_{\text{DPO}} &= \frac{\partial \mathcal{L}_{\text{DPO}}}{\partial a} \cdot \frac{\partial a}{\partial b} \cdot \frac{\partial b}{\partial \theta} \\ &= -\frac{1}{a} \cdot a \cdot (1-a)(-\beta T(\nabla_\theta \mathcal{L}_{\text{SFT}}(x_t^w) - \nabla_\theta \mathcal{L}_{\text{SFT}}(x_t^l))) \\ &= T\beta(1-a)(\nabla_\theta \mathcal{L}_{\text{SFT}}(x_t^w) - \nabla_\theta \mathcal{L}_{\text{SFT}}(x_t^l)), \end{aligned} \tag{23}$$

where,

$$a = \sigma\left[\beta T \log\left(\frac{p_\theta(x_{t-1}^w|x_t^w, c)}{p_{ref}(x_{t-1}^w|x_t^w, c)}\right) - \beta T \log\left(\frac{p_\theta(x_{t-1}^l|x_t^l, c)}{p_{ref}(x_{t-1}^l|x_t^l, c)}\right)\right] \tag{24}$$

$$b = \beta T \log\left(\frac{p_\theta(x_{t-1}^w|x_t^w, c)}{p_{ref}(x_{t-1}^w|x_t^w, c)}\right) - \beta T \log\left(\frac{p_\theta(x_{t-1}^l|x_t^l, c)}{p_{ref}(x_{t-1}^l|x_t^l, c)}\right) \tag{25}$$

$$= \beta T(\mathcal{L}_{\text{SFT}}(x_t^w) - \mathcal{L}_{\text{SFT}}(x_t^l)) - c \tag{26}$$

$$c = \beta T(\log p_{ref}(x_{t-1}^w|x_t^w, c) - \log p_{ref}(x_{t-1}^l|x_t^l, c)), \tag{27}$$

where $\mathcal{K}$ and $\mathcal{G}$ are the same as SFT (sec. A.3.1), represent as:

$$\mathcal{G}_{\theta,DPO}(x_t^o) = \Sigma^{-1}(x_t^o - \mu_\theta(x_t^o))$$
$$\mathcal{K}_\theta(x_t^o, x_t^w) = \langle \nabla_\theta \mu(x_t^o) \nabla_\theta \mu(x_t^w) \rangle, \tag{28}$$

As a result, the updating policy of DPO denotes as:

$$\Delta \log p_{\theta^{\tau+1}}(x_{t-1}^o | x_t^o) \propto \eta T \beta (1-a) \mathcal{G}_\theta(x_t^o) \cdot (\mathcal{K}_\theta(x_t^o, x_t^w) \mathcal{G}_\theta^T(x_t^w) - \mathcal{K}_\theta(x_t^o, x_t^l) \mathcal{G}_\theta^T(x_t^l)) + \mathcal{O}(\eta^2 \|\nabla \mathcal{L}_{DPO}\|^2). \tag{29}$$

For analysis, we define the core term within sigmoid function of the DPO objective as reward margin, shown in Eq. 30. $a$ is the function of reward margin for the win/lose pairs, due to the monotonicity of $\sigma(\cdot)$, a larger margin leads to larger $a$, which in turn restrains the updating strength. When the margin is negative, larger $\beta$ leads to a smaller $a$ and hence provides stronger updating step for the model to "catch up" the separating ability of the reference model faster. But when the model is better and has a positive margin, increasing $\beta$ will increase $a$ and hence create a negative influence on $\beta(1-a)$, which makes the model update less. This aligns well with the claims in original DPO paper: the stronger regularizing effect tends to "drag" current model back to ref model when its predictions deviate from ref too much.

$$\text{reward margin} = \frac{p_\theta(x_{t-1}^w | x_t^w, c)}{p_{ref}(x_{t-1}^w | x_t^w, c)} - \log \frac{p_\theta(x_{t-1}^l | x_t^l, c)}{p_{ref}(x_{t-1}^l | x_t^l, c)} \tag{30}$$

Similarly, we can extend the framework of updating policy for other DPO-style preference optimization methods, like Identity-preference Optimization (IPO, Azar et al. (2023)):

$$\mathcal{L}_{\text{IPO}} = -\min \left[ \left( \log \frac{p_\theta(x_{t-1}^w | x_t^w, c)}{p_{\text{ref}}(x_{t-1}^w | x_t^w, c)} - \log \frac{p_\theta(x_{t-1}^l | x_t^l, c)}{p_{\text{ref}}(x_{t-1}^l | x_t^l, c)} - \frac{1}{2\beta} \right)^2 \right],$$
$$\mathcal{G}_{IPO} = \mathcal{G}_{DPO}; \ a = \log \frac{p_\theta(x_{t-1}^w | x_t^w, c)}{p_{\text{ref}}(x_{t-1}^w | x_t^w, c)} - \log \frac{p_\theta(x_{t-1}^l | x_t^l, c)}{p_{\text{ref}}(x_{t-1}^l | x_t^l, c)} - \frac{1}{2\beta}. \tag{31}$$

For the Sequence Likelihood Calibration (SLiC, Zhao et al. (2022)), the updating policy becomes:

$$\mathcal{L}_{\text{SLiC}} = -\min \left[ \max \left[ 0, \delta - \log \frac{p_\theta(x_{t-1}^w | x_t^w, c)}{p_\theta(x_{t-1}^l | x_t^l, c)} \right] - \beta \log p_\theta(x_{t-1}^{\text{ref}} | x_t^{\text{ref}}, c) \right]$$
$$= \min \left[ \max \left[ 0, \delta + \mathcal{L}_{\text{SFT}}(x_t^w) - \mathcal{L}_{\text{SFT}}(x_t^l) \right] + \beta \mathcal{L}_{\text{SFT}}(x_t^{\text{ref}}) \right], \tag{32}$$
$$\mathcal{G}_{\text{SLiC}}(x_t^w) = \mathcal{A} \cdot \mathcal{G}_{\text{DPO}}(x_t^w) + \mathcal{G}_{\text{DPO}}(x_t^{\text{ref}}); \ \mathcal{A} = \mathbb{1} \left( \delta - \log \frac{p_\theta(x_{t-1}^w | x_t^w, c)}{p_\theta(x_{t-1}^l | x_t^l, c)} \right),$$

where $\mathbb{1}(\cdot)$ is the indicator function. In summary, these model-free based methods can all be calculated in the framework of updating policy. For the DPO and IPO loss, the directions of the updating signals are identical. A scalar controls the strength of this update, which usually correlates with the reward margin between chosen samples $x^w$ and rejected samples $x^l$. Generally, a larger reward margin leads to a larger $a$, making the norm of updating strength smaller. The SLiC loss can be considered as a combination of SFT adaptation and preference adaptation. If the reward margin is larger than $\delta$, the indicator function will become zero, and the model stops pushing chosen and rejected samples away when it already separates $x^w$ and $x^l$.

In summary, the updating policy framework we have proposed successfully elucidates the underlying update mechanisms of numerous fine-tuning methods, including SFT, DPO, and other DPO-style approaches such as IPO (Azar et al. (2023)) and SLiC (Zhao et al. (2022)). This unified perspective, in turn, provides crucial guidance for diagnosing and mitigating the known failure modes of the DPO method.

### A.3.3 Other Aspect of Likelihood Displacement

Next, we will explore the causes of likelihood displacement from the perspective of DPO's difference in the update range of chosen and rejected samples. According to the original paper of DPO (Rafailov et al. (2024)), its loss can be rewritten in the following form:

$$\mathcal{L}_{dpo} = -\min \left[ \log \ \sigma \left( \beta T \log \left( \frac{p_\theta(x_{t-1}^w | x_t^w, c)}{p_{ref}(x_{t-1}^w | x_t^w, c)} \right) - \beta T \log \left( \frac{p_\theta(x_{t-1}^l | x_t^l, c)}{p_{ref}(x_{t-1}^l | x_t^l, c)} \right) \right) \right]$$

$$= -\left[ \log \sigma \left( \frac{x_1^\beta}{x_1^\beta + x_2^\beta} \right) \right], \tag{33}$$

where $x_1 = \frac{p_\theta(x_{t-1}^w | x_t^w, c)}{p_{ref}(x_{t-1}^w | x_t^w, c)}$, $x_2 = \frac{p_\theta(x_{t-1}^l | x_t^l, c)}{p_{ref}(x_{t-1}^l | x_t^l, c)}$, $p(x^w) = p_\theta(x_{t-1}^w | x_t^w, c)$ and $p(x^l) = p_\theta(x_{t-1}^l | x_t^l, c)$ for simplification. In this case, we want to increase $x_1$ and decrease $x_2$ in DPO training.

The partial derivatives of Eq. 33 with respect to $x_1$ and $x_2$ are given as follows, detailed proof can be seen in previous work (Feng et al. (2024)).

$$\frac{\partial \mathcal{L}_{\text{DPO}}(x_1, x_2)}{\partial x_1} = -\frac{\beta x_2^\beta}{x_1(x_1^\beta + x_2^\beta)},$$

$$\frac{\partial \mathcal{L}_{\text{DPO}}(x_1, x_2)}{\partial x_2} = -\frac{\beta x_2^{\beta-1}}{x_1^\beta + x_2^\beta}. \tag{34}$$

This reveals an inherent asymmetry in the DPO update mechanism. For any given preference pair $(x^w, x^l)$, the DPO objective creates a stronger gradient signal to suppress the probability of the rejected sample, $p(x^l)$, than it does to amplify the probability of the chosen sample, $p(x^w)$. As training progresses and the reward margin increases, this imbalance becomes more pronounced. Consequently, the model's capacity to penalize the rejected sample is intrinsically greater than its capacity to reward the chosen one.

When this dynamic is viewed through the lens of our updating policy, a critical failure mode emerges. The model can exploit this asymmetry by taking a "path of least resistance" to minimize the loss. This seemingly counter-intuitive action involves reducing the probabilities of both the chosen and rejected samples simultaneously. This is possible because the downward "push" on $p(x^l)$ is stronger than the downward "pull" on $p(x^w)$. As a result, the log-probability gap, $\log(p(x^w)/p(x^l))$, can still increase—satisfying the optimization objective—even as both absolute probabilities degrade. This mechanism is the root cause of the "likelihood displacement" phenomenon, where the model improves relative preferences at the cost of declining absolute sample quality.

### A.4 Analysis of chosen probabilities enhancement

As illustrated in Eq. 10, we tracked the per-step log-likelihood advantage between PG-DPO and VideoDPO, demonstrating the enhancement of chosen sample probabilities. Based on Eq. 11,

$$\Gamma(\theta, \tau + 1) = \Delta \log p_{\theta^{\tau+1}}^{\text{Ours}}(x_{t-1}^w | x_t^w, c^w) - \Delta \log p_{\theta^{\tau+1}}^{\text{VideoDPO}}(x_{t-1}^w | x_t^w, c^w)$$

$$= -\eta \nabla_\theta \log p_{\theta^\tau}(x_{t-1}^w | x_t^w, c^w)(\nabla_\theta \mathcal{L}_{\text{Ours}}(x^w, x^l)|_{\theta^\tau} - \nabla_\theta \mathcal{L}_{\text{DPO}}(x^w, x^l)|_{\theta^\tau})^T. \tag{35}$$

**Detailed Derivation and Interpretation of Log-Likelihood Advantage.** To formally analyze the per-step improvement of our method, we start by defining the change in the log-likelihood of a chosen sample $x^w$ after a single gradient descent step. Using a first-order Taylor expansion for the function $f(\theta) = \log p_\theta(x^w | c)$ around the parameters $\theta^\tau$ at step $\tau$, we have:

$$\log p_{\theta^{\tau+1}}(x^w | c) \approx \log p_{\theta^\tau}(x^w | c) + \nabla_\theta \log p_{\theta^\tau}(x^w | c) \cdot (\theta^{\tau+1} - \theta^\tau). \tag{36}$$

The parameter update rule for gradient descent is $\theta^{\tau+1} - \theta^{\tau} = -\eta \nabla_\theta \mathcal{L}|_{\theta^\tau}$, where $\eta$ is the learning rate and $\mathcal{L}$ is the loss function. Substituting this into the expansion, the change in log-likelihood is:

$$\Delta \log p_{\theta^{\tau+1}}(x^w|c) = \log p_{\theta^{\tau+1}}(x^w|c) - \log p_{\theta^\tau}(x^w|c) \approx -\eta \nabla_\theta \log p_{\theta^\tau}(x^w|c) \cdot (\nabla_\theta \mathcal{L}|_{\theta^\tau})^T. \quad (37)$$

This equation quantifies how a single update step using a loss $\mathcal{L}$ affects the log-likelihood of the chosen sample. Our metric, $\Gamma(\theta, \tau + 1)$, is defined as the difference between the log-likelihood change induced by our method ($\mathcal{L}_{\text{Ours}}$) and that induced by the VideoDPO loss ($\mathcal{L}_{\text{VideoDPO}}$):

$$\Gamma(\theta, \tau + 1) = \Delta \log p_{\theta^{\tau+1}}^{\text{Ours}}(x^w|c) - \Delta \log p_{\theta^{\tau+1}}^{\text{VideoDPO}}(x^w|c). \quad (38)$$

By substituting the Taylor approximation for both terms, we arrive at the expression in Eq. 35, which provides a direct, step-by-step comparison of the two methods' impact on the chosen sample's probability.

**Why a Positive $\Gamma$ Indicates Superior Performance.** The core goal of preference alignment is to increase the model's likelihood of generating human-preferred (chosen) samples. The metric $\Gamma(\theta, \tau + 1)$ directly measures which training procedure is more effective at this goal on a per-step basis.

- **A Positive $\Gamma$:** If $\Gamma(\theta, \tau + 1) > 0$, it means that for the given preference pair $(x^w, x^l)$ at step $\tau$, the gradient update from our PG-DPO loss function provides a stronger "push" to increase the probability of $x^w$ compared to the update from the VideoDPO loss. This demonstrates that our method is more effectively steering the model parameters in the desired direction for the chosen sample.

- **Combating Likelihood Displacement:** The key failure mode we identified in DPO is "likelihood displacement", where the model may reduce the probability of both chosen and rejected samples to satisfy the loss. In such a scenario, $\Delta \log p^{\text{VideoDPO}}$ would be negative. A positive value for $\Gamma$ is particularly significant here, as it indicates that our method yields a less negative or more positive change in log-likelihood for $x^w$ than VideoDPO. It shows that PG-DPO is actively counteracting this undesirable behavior.

- **Interpreting the 81.7% and 76.9% Statistic:** Our empirical finding that $\Gamma$ is positive for 81.7% in Wanx2.1 model and 76.9% in VC2 model is a powerful statement about the *consistency* of our method's advantage. It implies that in more than four out of every five updates, PG-DPO is doing a better job than the baseline at the fundamental task of boosting the chosen sample's probability. This is not an averaged, end-of-training metric; it is a robust, step-by-step validation of our proposed update dynamics. This consistent, superior performance at the micro-level explains the superior alignment quality observed at the macro-level in our final model.

In summary, a positive $\Gamma$ serves as direct, quantitative evidence that PG-DPO provides a more effective and reliable learning signal for aligning with human preferences, successfully mitigating the critical failure modes inherent in the VideoDPO formulation.

## A.5 IMPLEMENTATION DETAILS

### A.5.1 METRIC DETAILS

- **CLIP Score:** To measure the semantic alignment between the generated video and the input text prompt, we compute the CLIP score for each frame and report the average value. This evaluates how well the video content matches the textual description.

- **HPS-v2:** To gauge how favorably humans would rate the generated videos, we employ HPS-v2 (Wu et al. (2023)), a reward model trained on large-scale human preference data. It predicts a score that correlates with human judgments on aesthetics and quality.

- **VideoAlign:** We utilize VideoAlign (Liu et al. (2025b)), a state-of-the-art evaluation model, to provide a holistic assessment of our generated videos. It scores them across multiple dimensions, including visual quality, motion quality, and text-video alignment.

- **Temporal Flickering:** To assess the temporal stability and color consistency, we calculate the temporal flickering across the video frames. A higher score indicates smoother transitions and fewer visual artifacts over time.

- **Aesthetic Quality:** We quantify the aesthetic appeal of the generated results using an established aesthetic predictor LAION-AI. (2022), which is trained on the LAION dataset to score visual aesthetics.

- **VQA Score:** To evaluate the objective quality and content fidelity of the generated videos, we leverage VQA model (Wu et al. (2022)). This approach assesses the visual quality within the video.

- **User Study:** In the user study, a total of 30 experienced participants are invited to take part. We employ PG-DPO (our method), SFT, DPO and baseline to separately generate 18 videos in both settings. As shown in Fig. 7a and Fig. 7b, users from diverse backgrounds evaluate the results in terms of visual quality (VQ), motion quality (MQ), text-video alignment (TA) and overall performance (Overall). Our method outperforms all state-of-art methods on four evaluation aspects with a big margin, demonstrating the effect of our method.

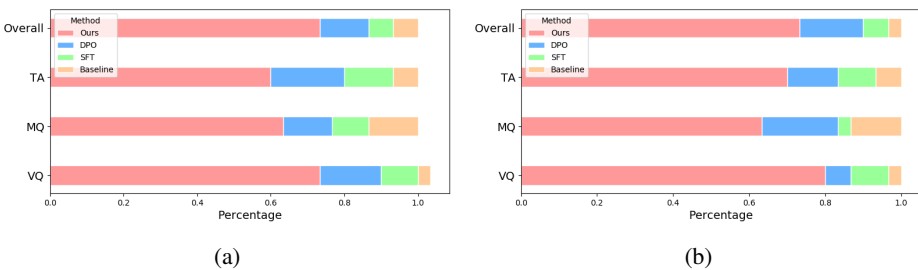

(a)  (b)

Figure 7: **Results for the user study in percentages.** (a) VideoCrafter2 (VC2), (b) Wanx2.1.

### A.5.2 MORE DETAILS OF FIG. 2

Figure 2 provides a micro-level visualization of the training dynamics, tracking the predicted probabilities of chosen and rejected samples. This analysis is crucial for empirically validating our claim that PG-DPO resolves the "likelihood displacement" failure mode inherent in VideoDPO.

**Probability Calculation in the Diffusion Model.** A key technical challenge is to define and compute a meaningful "probability" for a sample within a diffusion model framework like VideoCrafter-v2 (VC2), which utilizes a DDIM-like scheduler. Unlike autoregressive models that provide explicit next-token probabilities, the likelihood of a sample in a diffusion model is tied to the entire reverse denoising process. For our analysis, we employ a standard and computationally efficient proxy for the log-likelihood that is directly related to the model's training objective.

The reverse process in a diffusion model aims to denoise a variable $x_t$ to $x_{t-1}$. The transition probability $p_\theta(x_{t-1}|x_t, c)$ is modeled as a Gaussian distribution whose mean is a function of the model's noise prediction $\epsilon_\theta(x_t, t, c)$. We can use the model output to represent the mean of the Gaussian distribution, and then calculate the generation probability of a specific sample (Ho et al. (2020)).

**Analysis of Training Dynamics.** With this methodology, the plots in Figure 2 become clear.

- **Fig. 2(a) - VideoDPO:** This plot exhibits the classic symptoms of likelihood displacement. The dashed orange line (Rejected) shows a clear downward trend, as the DPO loss successfully penalizes the rejected sample. However, the solid blue line (Chosen) also degrades over time, or at best, stagnates. This occurs because DPO's update mechanism, focused solely on maximizing the reward margin, often finds it easier to suppress both probabilities, especially when the chosen and rejected samples are similar. While the difference (lower subplot) may slowly increase, this comes at the cost of the model's absolute ability to generate the preferred sample, leading to a suboptimal alignment.

- **Fig. 2(b) - Our Method (PG-DPO):** In stark contrast, our method demonstrates a much healthier alignment process. The probability of the chosen sample (solid blue line) shows a consistent and stable upward trend throughout training. Simultaneously, the probability of the rejected sample (dashed orange line) is effectively suppressed. This is a direct consequence of our design: the adaptive regularization term controlled by $\gamma$ provides a direct incentive to increase the chosen sample's likelihood, while the trade-off parameter $\alpha$ prevents the penalty on the rejected sample from negatively impacting the chosen one. The result is a model that not only learns the preference (as shown by the steadily increasing difference in the lower subplot) but also improves its fundamental capability to generate high-quality, preferred content.

In conclusion, this detailed analysis confirms that PG-DPO successfully addresses the underlying cause of DPO's likelihood displacement by ensuring the probability of desired outputs is consistently enhanced, rather than just optimizing a relative margin at the expense of absolute quality.

### A.5.3 ANALYSIS OF HYPER-PARAMETERS

Our proposed PG-DPO method introduces two key hyperparameters, $K_1$ and $K_2$. In this section, we provide an ablation study on their effects during training and offer rationales for their selection, substantiating our choices with empirical evidence.

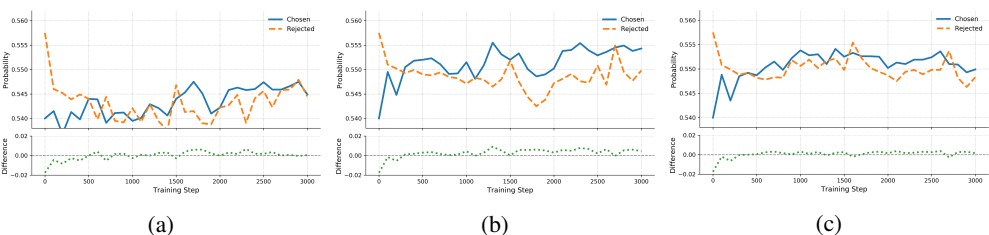

(a)          (b)          (c)

Figure 8: **Predicted probabilities for chosen and rejected samples across training steps under different $K_1$.** (a) $K_1 = 2, \alpha \in [0.5, 0.6)$. (b) $K_1 = 10, \alpha \in [0.5, 0.8)$. (c) $K_1 = 20, \alpha \in [0.5, 1)$.

**Effect of $K_1$**    The hyperparameter $K_1$ directly controls the value of $\alpha$. A smaller $K_1$ results in a smaller $\alpha$. When $\alpha$ is excessively small (e.g., in the range $[0.5, 0.6)$ as shown in Fig. 8a), it leads to an overly aggressive suppression of the rejected samples. Consequently, the model struggles to effectively increase the probability margin between chosen and rejected samples, as the growth of the chosen probability is hindered. Conversely, an overly large $\alpha$ (e.g., in the range $[0.5, 1)$ as seen in Fig. 8c) leads to insufficient suppression of the rejected samples. This is particularly problematic when the reward margin is small, as it hinders the desired shift in the chosen distribution. In extreme cases, this can lead to "likelihood displacement", where the probabilities of both chosen and rejected samples decrease simultaneously. Therefore, the selection of $K_1$ involves a trade-off. We recommend dynamically tuning $K_1$ based on the specific dataset, with the core objective of maintaining $\alpha$ within a stable and effective range, such as $[0.5, 0.8)$, which demonstrates a clear and consistent separation between sample probabilities (Fig. 8b).

**Effect of $K_2$**    The hyperparameter $K_2$ influences both the value and the operational range of $\gamma$. A larger $K_2$ yields a lower average value for $\gamma$. This, in turn, diminishes the relative weight of the implicit reward regularization for chosen samples within the overall loss function. As depicted in Fig. 9a, when $K_2$ is large (leading to $\gamma \in [0.2, 0.5)$), the regularization term has a minimal impact even for samples with a large reward margin. This makes it difficult to overcome the "suboptimal maximization" problem, manifesting as the model's inability to consistently increase the probability of chosen samples as training progresses.Conversely, if $K_2$ is too large, the PG-DPO loss degenerates into the SFT loss. A detailed comparison and empirical analysis of this scenario are provided in Section 4. Thus, selecting an appropriate $K_2$ is crucial. It should be tuned according to dataset characteristics to ensure that $\gamma$ operates within a meaningful range (e.g., $(0, 0.5)$ as shown in Fig. 9b), thereby balancing preference optimization with effective regularization.

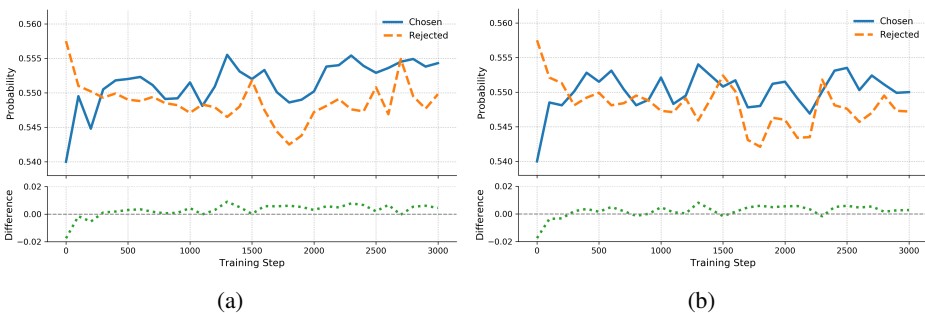

Figure 9: **Predicted probabilities for chosen and rejected samples across training steps under different** $K_2$. (a) $K_2 = 10, \gamma \in (0, 0.5)$. (b) $K_2 = 5, \gamma \in [0.2, 0.5]$.

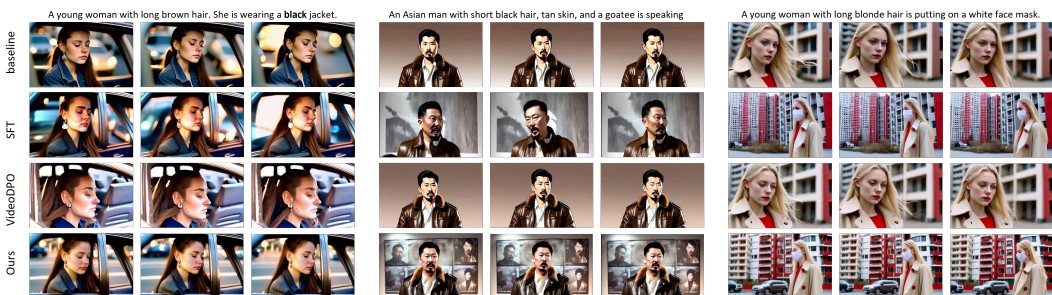

Figure 10: Comparison results with previous works on the VideoCrafter (VC2).

**Effect of** $\beta$. The parameter $\beta$ governs the trade-off between maximizing the learned preference and remaining faithful to the reference model's distribution. A higher $\beta$ prioritizes fidelity to the reference, which is particularly useful for mitigating instability during training. This mechanism, as previously detailed, acts as a safeguard against catastrophic deviations that can lead to quality degradation. Consequently, for scenarios where outputs become unstable or corrupted, a larger $\beta$ should be employed. Across our video generation experiments, we found values in the range of $[100, 2000]$ to be effective.

## A.5.4 CORE CODE OF PG-DPO

In this section, we present the core implementation of the proposed PG-DPO method. This code can be seamlessly integrated into any video generation pipeline, with the adaptive parameter adjustment scheme detailed in the preceding discussion.

```python
def dpo_loss_fn(model, model_ref, latents, target, t, beta, K1, K2):
    """
    Computes the PG-DPO loss.

    Args:
    - model: The policy model being trained.
    - model_ref: The frozen reference model.
    - latents: Batched inputs for both chosen and rejected samples.
    - target: The ground truth noise for the chosen and rejected
        samples.
    - t: The current diffusion timestep.
    - beta: The temperature parameter controlling the deviation from
        the reference model.
    - K1, K2: Sensitivity parameters for the adaptive weighting
        functions.

    Returns:
    - loss: The final PG-DPO loss value.
    """
```

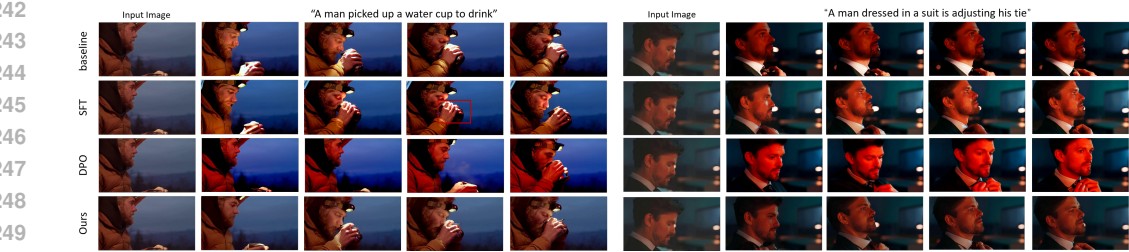

Figure 11: Comparison results with previous works on the Wanx2.1.

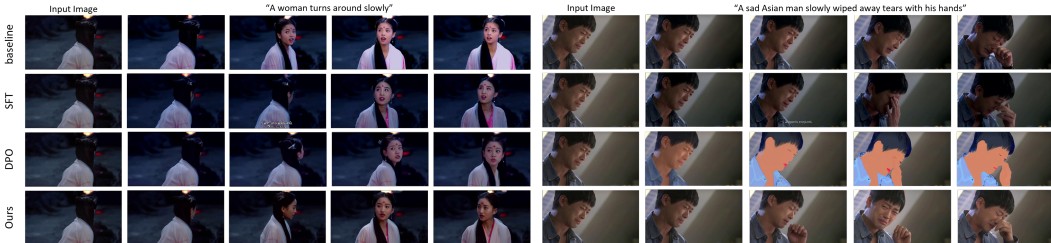

Figure 12: Comparison results with previous works on the Wanx2.1.

```
17    pred = model(latents, t)
18    ref_pred = model_ref(latents, t)
19
20    w_pred, l_pred = pred.chunk(2)
21    ref_w_pred, ref_l_pred = ref_pred.chunk(2)
22    reduce_dims = list(range(1, w_pred.ndim))
23
24    w_err = (w_pred - w_target).pow(2)
25    l_err = (l_pred - l_target).pow(2)
26    ref_w_err = (ref_w_pred - w_target).pow(2)
27    ref_l_err = (ref_l_pred - l_target).pow(2)
28
29    w_diff = w_err - ref_w_err
30    l_diff = l_err - ref_l_err
31
32    # Alpha: to down-weight the penalty on the rejected sample
33    alpha = torch.sigmoid(K1 * ((w_diff - l_diff) / (l_diff + 1e-6)))
34
35    # Gamma: to switch between DPO and regularization.
36    gamma = torch.sigmoid(-K2 * (w_diff - l_diff) / (l_diff + 1e-6)))
37
38    # Core DPO objective, modulated by alpha
39    inside_term = -0.5 * beta * (w_diff - l_diff * alpha)
40    loss_DPO = - torch.nn.functional.logsigmoid(inside_term).mean(dim=
          reduce_dims)
41
42    # IPR term for stability
43    # Encourages the policy model to improve on chosen samples w.r.t
          the reference
44    loss_IPR = ((w_pred - w_target) - (w_ref_pred - w_target)).pow(2).
          mean(dim=reduce_dims)
45
46    # Final loss is a dynamic interpolation between DPO and IPR,
          controlled by gamma
47    loss = gamma * loss_DPO + (1 - gamma) * loss_IPR
48
49    return loss
```

## A.6 MORE RESULTS

As demonstrated by the qualitative results in Fig. 10, Fig. 11, and Fig. 12, our method exhibits clear advantages over previous works. Specifically, our approach excels at enhancing the visual texture and aesthetic quality of the generated videos. Moreover, it effectively mitigates undesirable motion artifacts and distortions, leading to more coherent and realistic movements. Finally, the alignment between the video content and the text prompt is also substantially improved, demonstrating a more robust semantic understanding.

