# OpenReview forum: "Beyond Reward Margin: Rethinking and Resolving Likelihood Displacement in Diffusion Models via Video Generation"
_ICLR.cc/2026/Conference — ICLR 2026 Conference Desk Rejected Submission_

### Official Review · Reviewer_VfvG · 2025-10-16

**Soundness:** 2
**Presentation:** 3
**Contribution:** 2
**Rating:** 4
**Confidence:** 3

**Summary:**

This paper analyzes likelihood displacement in DPO for diffusion-based video generation, identifying two failure modes and proposing PG-DPO with Adaptive Rejection Scaling and Implicit Preference Regularization to resolve them. The method is empirically validated, and shows consistent improvements, though hyperparameter sensitivity and limited human evaluation leave room for improvement.

**Strengths:**

**Clarity:**

1. The paper is well-structured: problem → analysis → solution → experiments.

2. Visualizations (likelihood trajectories, qualitative generations) clearly support the claims.

**Weaknesses:**

**Hyperparameter sensitivity:**
PG-DPO introduces new hyperparameters. The paper admits these require careful tuning, but no adaptive scheme or systematic guideline is provided. Ablation study is encouraged.

**Generality claims:**
The framework is said to be extensible to various fine-tuning methods, but no empirical validation outside DPO is provided.

**Questions:**

1. Hyperparameter sensitivity. See W1.

2. Does the regularization have similar effect with MaPPO [1]? Some discussion of similar approaches are encouraged.

[1] MaPPO: Maximum a Posteriori Preference Optimization with Prior Knowledge. arXiv:2507.21183, 2025.

---

### Official Review · Reviewer_c9mr · 2025-10-30

**Soundness:** 2
**Presentation:** 3
**Contribution:** 2
**Rating:** 4
**Confidence:** 3

**Summary:**

This paper presents PG-DPO, by modifying DPO objective with adaptive rejection scaling and implicit reference regularization. The initiative is based on two failure modes of DPO: optimization conflict and suboptimal maximization. The proposed method is applied to preference alignment of video generation, specifically VideoCrafter2 and Wanx2.1. The PG-DPO shows improved metrics compared with SFT and VideoDPO.

**Strengths:**

- Modify DPO objective based on identified failure modes.

**Weaknesses:**

- The modification to DPO is considered as incremental.
- Insufficient experiments:
  - The proposed method is not validated on image generation.
  - The proposed method is only compared with VideoDPO in experiments.

**Questions:**

- How's this method work on text-to-image generation?

---

### Official Review · Reviewer_Tc72 · 2025-11-09

**Soundness:** 3
**Presentation:** 3
**Contribution:** 3
**Rating:** 4
**Confidence:** 3

**Summary:**

The paper provides the analysis of likelihood displacement in diffusion models and introduces Policy-Guided DPO (PG-DPO) — a theoretically grounded, empirically validated method that stabilizes preference alignment for video generation.

**Strengths:**

1. The paper provides a novel formal decomposition of DPO's updating dynamics in diffusion models, revealing actionable failure modes.

2. The proposed approach PG-DPO effectively combines ARS and IPR to address both small- and large-margin issues, with empirical evidence (e.g., Fig. 2) showing consistent probability increases for chosen samples.

3. Also the framework extends to other fine-tuning algorithms (e.g., SFT, KTO) and high-dimensional tasks like video generation.

**Weaknesses:**

1. Hyperparameters (e.g., K1, K2 in ARS/IPR) introduce tuning complexity without clear ablation studies.

2. Experimental details (e.g., datasets, baselines, quantitative metrics) are referenced but not fully provided in the visible pages, making it hard to assess reproducibility or superiority claims. for example, SFT is proven to be the most effective way for post-training. The paper lacks the pipeline of choosing post-training data. Is the DPO done only on pretraining or also on SFT as well? it is not very clear.

**Questions:**

1. How was the preference dataset collected (e.g., human annotators, synthetic pairs), and what scale was used for training/evaluation?

2. Can you provide more details on the experimental setup, including baselines (e.g., VideoDPO, Diffusion-DPO), metrics (e.g., FVD, human evaluations), and computational resources?

3. Does PG-DPO generalize to non-video diffusion tasks, such as text-to-image, and were any such experiments conducted?

4. Were ablation studies performed to isolate the impacts of ARS and IPR, and how sensitive is performance to hyperparameters like K1, K2, and β?

5. How does PG-DPO compare to recent DPO variants (e.g., SPIN-Diffusion) in terms of stability and preference data efficiency?

---

### Note · Program_Chairs · 2026-01-17
**Submission Desk Rejected by Program Chairs**

The following references in this submission do not refer to real documents and/or have major errors in bibliographic information:

 Yifan Pu, Jihan Liu, Feifan Wang, Long-Kai Ma, Zhipeng Wang, Zilong Li, Tianyu Xue, Jiazheng Liu, Zihan Zhao, Hanting Lu, Xiaoguang Wang, and Saining Song. RLHF-V: Towards general-purpose video-language model via reward-guided tuning. arXiv preprint arXiv:2402.16423, 2024.